# Entropic Matroids and Their Representation

**DOI:** 10.3390/e21100948

**Published:** 2019-09-27

**Authors:** Emmanuel Abbe, Sophie Spirkl

**Affiliations:** 1Department of Mathematics, École polytechnique fédérale de Lausanne, 1015 Lausanne, Switzerland; 2Department of Mathematics, Princeton University, Princeton, NJ 08540, USA; sspirkl@math.princeton.edu

**Keywords:** matroids, entropy function, extremal dependencies, combinatorics, coding

## Abstract

This paper investigates *entropic matroids*, that is, matroids whose rank function is given as the Shannon entropy of random variables. In particular, we consider *p-entropic matroids*, for which the random variables each have support of cardinality *p*. We draw connections between such entropic matroids and secret-sharing matroids and show that entropic matroids are linear matroids when p=2,3 but not when p=9. Our results leave open the possibility for *p*-entropic matroids to be linear whenever *p* is prime, with particular cases proved here. Applications of entropic matroids to coding theory and cryptography are also discussed.

## 1. Introduction

Matroid theory generalizes the notion of independence and rank beyond vector spaces. In a graphical matroid, for example, the rank of a subset of edges is the size of an acyclic spanning set of edges; analogous to the rank of a subset of vectors, which is the size of a spanning set of linearly independent vectors. It is natural to ask whether such combinatorial structures can also be obtained from probabilistic notions of independence, based on random variables. In particular, the entropy can be used to measure dependencies between random variables and it can be used to define a matroid rank function as discussed below. One can then investigate how such entropic matroids relate to other matroids, in particular whether they admit linear representations as graphical matroids do. Before giving formal definitions of such entropic matroids, we give some general definitions for matroids.

### 1.1. Definitions

We recall a few standard definitions related to matroids, see, for example, Oxley [1]. A *matroid* is a pair M=(E,r), where the ground set *E* is a finite set (typically E=[m], $m\in Z_{+}$) and where the rank function $r:2^{E}\to Z_{+}$ satisfies
For any A⊆E, r(A)≤|A| (*normalization*);For any A⊆B⊆E, r(A)≤r(B) (*monotonicity*);For any A,B⊆E, r(A∪B)+r(A∩B)≤r(A)+r(B) (*submodularity*).

The submodularity property can be interpreted as a diminishing return property: for every A⊆B and x∈E,
(1)$$\begin{array}{c} r(A\cup x)-r(A)\geq r(B\cup x)-r(B), \end{array}$$
that is, the larger the set, the smaller the increase in rank when adding a new element. Independent sets in a matroid are the subsets S⊆E such that r(S)=|S| and maximal independent sets are called *bases*, whereas minimal dependent sets are called *circuits*.

A matroid M=(E,r) is *linear* if there is a vector space *V* and a map f:E→V such that r(S)=rank(f(S)) for all S⊆E, where rank denotes the rank function of *V*, that is, rank(f(S))=dimspan(f(S)). We say that a matroid is *F-representable* if in addition, *V* can be chosen as a vector space over the field F.

Given a matroid *M*, a *minor* of M=(E,F) is a matroid that can be obtained from *M* by a finite sequence of the following two operations:*Restriction*: Given A⊆E, we define the matroid $M|A=(A,F\cap 2^{A})$.*Contraction*: Given an independent set A∈F, we define the matroid $M/A=(E\setminus A,\left\{B\subseteq E\setminus A:B\cup A\in F\right\})$.

We define the *dual*
$M^{*}=\left(E,r^{*}\right)$ of a matroid M=(E,r) is defined by letting $r^{*}\left(A\right)=r\left(E\setminus A\right)+\left|A\right|-r\left(E\right)$ for all A⊆E. A matroid property is a *dual property* if *M* has the property if and only if $M^{*}$ does.

**Theorem** **1**(Woodall [2]). *Being an F-representable matroid is a dual property, thas is, M is F-representable if and only if $M^{*}$ is.*

### 1.2. Entropic Matroids

One may expect that matroids could also result from probabilistic structures. Perhaps the first possibility would be to define a matroid to be ‘probabilistic’ if its elements can be represented by random variables (with a joint distribution on some domain), such that a subset *S* is independent if the random variables indexed by *S* are mutually independent. This, however, does not necessarily give a matroid. For example, let $X_{1}$ and $X_{2}$ be independent random variables (for example, normally distributed) and let $X_{3}=X_{1}+X_{2}$. Let A={3}, B={1,3} and x={2}. Then r(A∪x)−r(A)=0 since $X_{2}$ and $X_{3}$ are dependent but r(B∪x)−r(B)=1 since B∪x={1,2,3} contains two independent random variables. So this violates the submodularity requirement.

On the other hand, it is well known that the entropy function satisfies the monotonicity and submodularity properties [3,4]. Namely, for a probability measure μ on a discrete set X, the *entropy of μ in base q* is defined by
(2)$$\begin{array}{cc} H(\mu ) & =-\sum_{x\in X}\mu \left(x\right)\log_{q}\mu \left(x\right). \end{array}$$

For two random variables *X* and *Y* with values in X and Y respectively and with joint distribution μ, we define the *conditional entropy*
(3)$$\begin{array}{c} H\left(X|Y\right)=\sum_{x\in X,y\in Y}\mu \left(x,y\right)\log\frac{\mu (x,y)}{\sum_{u\in X}\mu \left(u,y\right)}. \end{array}$$

In particular, we have the chain rule of entropy H(X|Y)=H(X,Y)−H(Y). We also define the *Hamming distance* of two vectors *x* and *y* as $d\left(x,y\right)=|\left\{1\leq i\leq n:x_{i}\neq y_{i}\right\}|$ and the *Hamming ball* of radius *r* around *x* as $B_{r}\left(x\right)=\left\{y:d(x,y)\leq r\right\}$.

Furthermore, for a probability measure μ of *m* random variables defined each on a domain X, that is, for a probability distribution μ on $X^{m}$, one can define the function
(4)$$\begin{array}{c} r\left(S\right)=H\left(\mu_{S}\right),\;S\subseteq \left[m\right], \end{array}$$
where $\mu_{S}$ is the *marginal* of μ on *S*, that is,
(5)$$\begin{array}{cc} \mu_{S}\left(x\left[S\right]\right) & =\sum_{x_{i}\in \left[q\right]:i\notin S}\mu \left(x\right),\;x\left[S\right]=\left\{x_{i}:i\in S\right\}. \end{array}$$

By choosing the base *q* for the entropy in (Equation 4) to be |X|, we also get that r(S)≤|S|, with equality for uniform measures. Therefore, the above *r* satisfies the three axioms of a rank function, with the exception that *r* is not necessarily integral. In fact this defines a polymatroid (and *r* is also called a β-function [5]) and entropic polymatroids (i.e., polymatroids derived from such entropic β-functions) have been studied extensively in the literature; see References [6,7,8,9] and references therein. Using the Shannon entropy to study matroid structures already emerged in the works [10,11], where the family of pairs of sets (i,j) and *K* such that K⊆[m], i,j∈[m]∖K is called probabilistically representable if there exit random variables ${\left\{X_{k}\right\}}_{k\in [m]}$ such that $X_{i}$ and $X_{j}$ are conditionally independent given $X_{K}$, with the latter expressed in terms of the Shannon entropy as r(i,K)+r(j,K)−r(i,j,K)−r(K)=0.

However, we can also investigate what happens *if* this function *r* is in fact integral. This is the object of study in this paper.

**Definition** **1.***Let $q\in Z_{+}$. A matroid M=([m],r) is q*-entropic*if there is a probability distribution μ on ${\left[q\right]}^{m}$ such that for any S⊆[m],*(6)$$\begin{array}{c} r\left(S\right)=H\left(\mu_{S}\right), \end{array}$$*where $\mu_{S}$ is the marginal of μ on S and H is the Shannon entropy in base q.*

Note that the entropy does not depend on the support of the random variables but only on their joint distribution. For this reason, the restriction that μ is taking values in ${\left[q\right]}^{m}$ is in fact equivalent to requiring that each random variable has a support of cardinality at most *q*. When working with the *m* underlying random variables $X_{1},\cdots ,X_{m}$ distributed according to μ, we write $H\left(S\right)=H\left(X\left[S\right]\right)=H\left(X_{i}:i\in S\right)=H\left(\mu_{S}\right)$.

With the integrality constraint, the random variables representing a *q*-entropic matroid must be marginally either uniformly distributed or deterministic, each pair of random variables must be either independent or a deterministic function of each other, and so on. These represent therefore extremal dependencies. As discussed in Section 8, such distributions (with extremal dependencies) have recently emerged in the context of polarization theory and multi-user polar codes [12], which has motivated in part this paper. In Section 4, we also comment on the connection between secret sharing from cryptography.

It is well-known and easy to check that entropic matroids generalize linear matroids, see, for example, References [7,13]. For completeness we recall the proof, making explicit the dependency on the field size.

**Lemma** **1.**
*Let F be a finite field. If a matroid is F-representable then it is |F|-entropic.*


**Proof.** Let *M* be an F-representable matroid and *A* be a matrix in $F^{|E|\times n}$ whose rows correspond to elements of *E* so that a subset of rows is linearly independent in $F^{n}$ if and only if the corresponding subset of *E* is independent in *M*. Let $Y_{1},\cdots ,Y_{n}$ be mutually independent and uniformly distributed random variables over F and let $Y=(Y_{1},\cdots ,Y_{n})$. Then the vector of random variables $(X_{1},\cdots ,X_{\left|E\right|})=A\cdot Y$ satisfies that for any B⊆E, $H\left(\left\{X_{i}:i\in B\right\}\right)=\mathrm{rank}\left\{A_{i}:i\in B\right\}$. Thus the entropy function on $X_{1},\cdots ,X_{\left|E\right|}$ recovers the rank function of *M* and *M* is |F|-entropic. □

Our main goal throughout the remainder of this paper is to investigate whether entropic matroids are always representable over fields. As discussed in next section, we will approach this question by checking whether the forbidden minors of representable matroids are entropic or not. This strategy is justified by the fact that for the Shannon entropy, entropic matroids are a minor-closed class, as we will show in Lemma 2.

### 1.3. Results

We prove that for every *p*, a matroid is *p*-entropic if and only if it is secret-sharing with a ground set of size *p*, which is equivalent to being the matroid of an almost affine code with alphabet size *p*. Furthermore, we prove that for every *p*, being *p*-entropic is closed under taking matroid minors.

We give alternative proofs that for p=2 and p=3, being *p*-entropic is equivalent to being $F_{p}$-representable by examining known forbidden minor characterizations. We also make some partial progress towards proving the same for other primes *p*. In the final section of the paper, we mention some applications of entropic matroids in coding.

## 2. Further Related Literature

Matroid representations and forbidden minors were studied in Reference [14] for GF(3), Reference [15,16] for GF(4) and some results for general fields were obtained in References [17,18,19]. Linear representable matroids are also intimately related to linear solutions to network coding problems, in particular in Reference [20], in which a network-constrained matroid enumeration algorithm is developed, as well as Reference [21] that considers integer-valued polymatroids and representable polymatroids in References [22,23]. Matroid’s minors and the connection to Zhang-Yeung inequality was discussed in Reference [24], which shows in particular that almost entropic matroids have infinitely many excluded minor. Matroids, secret sharing and linearity are also discussed in several papers as mentioned in part earlier. Reference [25] gave the first example of an access structure (i.e., the parties that can recover the secret from their share) induced by a matroid, namely the Vamos matroid, that is non-ideal (a measure of optimality of the secret shares lengths); Reference [26] presented the first non-trivial lower bounds on the size of the domain of the shares for secret-sharing schemes realizing an access structure induced by the Vamos matroid and this is later improved in Reference [27] using using non-Shannon inequalities for the entropy function. As mentioned earlier, an important line of work is also dedicated to understanding the representation of entropic polymatroids for a fixed ground set cardinality [9], which is well-understood for cardinality 2 and 3 and more complicated for larger cardinality with the non-Shannon inequalities emerging.

## 3. Minors of Entropic Matroids

In this section, we prove the following:

**Lemma** **2.**
*Let M be an entropic matroid on random variables $X_{1},\cdots ,X_{m}$ with values in $F_{p}$ and with entropy H and joint distribution μ.*
*(i)* 
*For any $A\subseteq \left\{X_{1},\cdots ,X_{m}\right\}$, M|A is entropic.*
*(ii)* 
*For any $X_{i}\in \left\{X_{1},\cdots ,X_{m}\right\}$ with $H(X_{i})=1$, $M/\left\{X_{i}\right\}$ is entropic.*
*(iii)* 
*For any independent set A, M/A is entropic.*



**Proof.** For each of the claims, we construct random variables and a probability distribution whose entropy agrees with the rank function of the matroid in question.To prove (i), we consider the variable set *A* with the marginal distribution given by μ. Then *H* is integral on any subset of *A*, since it is integral on any subset of $\left\{X_{1},\cdots ,X_{m}\right\}$. This implies (i).To prove (ii), we consider two cases. If for any $B\subseteq \left\{X_{1},\cdots ,X_{m}\right\}$ with $X_{i}\notin B$ we have $H\left(X_{i},B\right)=H\left(B\right)+1$, then $X_{i}$ is independent of all other variables. In particular, any set is independent in *M* if any only if its union with $\left\{X_{i}\right\}$ is. Therefore, $M/\left\{X_{i}\right\}=M|\left\{X_{1},\cdots ,X_{i-1},X_{i+1},\cdots ,X_{m}\right\}$ in this case and the result follows from (i).Otherwise, we define a distribution on $\left\{X_{1},\cdots ,X_{i-1},X_{i+1},\cdots ,X_{m}\right\}$ by fixing any value *x* for $X_{i}$ with $P\left[X_{i}=x\right]>0$ and considering the probability distribution obtained by conditioning on the event $\left\{X_{i}=x\right\}$. Now let $A\subseteq \left\{X_{1},\cdots ,X_{i-1},X_{i+1},\cdots ,X_{m}\right\}$. There are two cases. If there is no circuit *C* with $X_{i}\in C$ such that *A* contains $C\setminus \left\{X_{i}\right\}$ as a subset, then $H\left(A\right)+1=H\left(A,X_{i}\right)=H\left(A\right)+H\left(X_{i}|A\right)$, therefore $H(X_{i}|A)=1$ and so $X_{i}$ and *A* are independent. In this case, $H\left(A|X_{i}=x\right)=H\left(A\right)$, thus *H* agrees with the rank function of $M/\left\{X_{i}\right\}$.If adding $X_{i}$ to *A* creates a circuit, then $H\left(A,X_{i}\right)=H\left(A\right)$ and $H\left(A|X_{i}\right)=H\left(A\right)-1$. Let X(A) denote the vector with components $X_{j},j\in A$ and let $Y=F_{p}^{A}$ denote the set of possible values of X(A).Suppose first that $H\left(A|X_{i}=k\right)<H\left(A\right)-1$ for some k∈F. Now let *B* be a basis in *A*, that is, |B|=H(B)=H(A). We have that $H\left(A|X_{i}=k\right)=H\left(B|X_{i}=k\right)+H\left(A|B,X_{i}=k\right)$ and $H\left(A|B,X_{i}=k\right)\leq H\left(A,X_{i}|B\right)=H\left(A|B\right)=0$. Therefore, $H(B|X_{i}=k)<|B|-1$.Now let *C* be the unique circuit in B∪{i}. It follows that $H\left(C\right)=H\left(C\setminus \left\{X_{i}\right\}\right)=\left|C\right|-1$ and H(B∖C|C)=H(B)−H(C)=|B∖C|. In particular, the variables in B∖C are independent of $X_{i}$ in the marginal distribution on *B* and thus
$$H\left(B|X_{i}=k\right)=H\left(B\setminus C\right)+H\left(C\setminus \left\{X_{i}\right\}|X_{i}=k,B\setminus C\right)=\left|B\setminus C\right|+H\left(C|X_{i}=k\right).$$This implies that $H(C|X_{i}=k)<|B|-|B\setminus C|-1=|C|-2$. But $P\left[X_{i}=k|X\left(C\setminus \left\{X\right\}\right)=c\right]\in \left\{0,1\right\}$ and $P\left[X(C\setminus \left\{X_{i}\right\})=c\right]=p^{-|C|+1}$, which implies that $P\left[X(C)=c\right]\in \left\{0,p^{-|C|+1}\right\}$ and $P\left[X\left(C\setminus \left\{X_{i}\right\}\right)=c|X_{i}=k\right]\in \left\{0,p^{-|C|+2}\right\}$. Since these probabilities add up to one, it follows that exactly $p^{|C|-2}$ of them are non-zero, which yields
$$\begin{array}{cc} H(C|X_{i}=k) & =\sum_{c}P\left[X\left(C\setminus \left\{X_{i}\right\}\right)=c|X_{i}=k\right]\log_{p}\left(\frac{1}{P\left[X\left(C\setminus \left\{X_{i}\right\}\right)=c|X_{i}=k\right]}\right).\\ \; & =p^{|C|-2}\left(p^{-|C|+2}\log_{p}\left(\frac{1}{p^{-|C|+2}}\right)\right)\\ \; & =|C|-2, \end{array}$$
a contradiction to the assumption H(C)<|C|−2.This implies that $H\left(A|X_{i}=k\right)\geq H\left(A\right)-1$ for all *A*. Since
$$\begin{array}{cc} H(A)-1 & =H\left(A|X_{i}\right)=\sum_{k=0}^{p-1}P\left[X_{i}=k\right]H\left(A|X_{i}=k\right)\\ \; & =\sum_{k=0}^{p-1}\frac{1}{p}H\left(A|X_{i}=k\right)\geq p\cdot \frac{1}{p}\left(H\left(A\right)-1\right)=H\left(A\right)-1, \end{array}$$
it follows that we have $H\left(A|X_{i}=k\right)=H\left(A\right)-1$ for all summands. This implies that the entropy of the conditional distribution yields the entropic matroid $M/\left\{X_{i}\right\}$ and this proves (ii).Finally, (iii) follows by applying (ii) repeatedly. □

This lemma proves that the property of being an entropic matroid is closed under taking minors. This means that in order to show entropic matroids belong to a minor-closed class of matroids, it suffices to show that the forbidden minors of this class are not entropic.

## 4. Secret-Sharing and Almost Affine Matroids

Secret-sharing matroids were introduced in Reference [28]. These matroids are motivated by the problem of secret-sharing in cryptography [29,30], which refers to distributing a secret among a collection of parties via secret shares such that the secret can be reconstructed by combining a sufficient number (of possibly different types) of secrete shares, while individual shares being of no use on their own.

We use the following definitions from Reference [25]: Let $A\in S^{I\times E}$ be a matrix, where S,I and *E* are finite sets. For i∈I, e∈E and $Y\subseteq E\setminus \left\{e\right\}$, we define $n\left(i,e,Y\right)=\left\{a_{je}:j\in I,a_{jy}=a_{iy}forally\in Y\right\}$. Then *A* is a *secret-sharing matrix* if for e∈E and $Y\subseteq E\setminus \left\{e\right\}$, either n(i,e,Y)=S for all i∈I or |n(i,e,Y)|=1 for all i∈I. Any secret-sharing matrix induces a *secret-sharing matroid* with ground set *E* and rank function r(Y) the logarithm with base |S| of the number of distinct rows of the submatrix $A\left[Y\right]=\left(a_{ij}:i\in I,j\in Y\right)$ of *A*. In particular, *Y* is independent if and only if A[Y] contains all vectors in $S^{Y}$.

The interpretation is as follows. Suppose some row i∈I has been chosen in *A* but its value has been kept secret. Knowing *A*, one wishes to determine as much as possible about the values $a_{ie},e\in E$, without knowing which row has been selected. If by some means one has been able to determine the values $a_{if}$ for all f∈Y⊆E. Then the possible values of $a_{ie}$ for some e∈E∖Y, consistent with the available information, are precisely the members of n(i,e,Y) (and this set can be determined despite not knowing *i*).

Secret-sharing matroids were connected to entropy rank functions in Reference [31], as further discussed below. We now formally connect the two classes of matroids.

**Lemma** **3.**
*If a matroid is p-entropic, then it is a secret-sharing matroid with a ground set of size p.*


**Proof.** Given a *p*-entropic matroid *M* with ground set *E* and rank (entropy) function *H*, we let *A* be the matrix containing all vectors in $Z_{p}^{E}$ which correspond to outcomes of positive probability in *M*. For every set *Y* of variables, A[Y] contains the possible outcomes of these variables. These outcomes are all equally likely and the number of distinct outcomes with positive probability is $p^{H(Y)}$. This implies that to prove that *M* is a secret-sharing matroid, it suffices to prove that *A* is a secret-sharing matrix.Let e∈E and $Y\subseteq E\setminus \left\{e\right\}$. Then n(i,e,Y) is the number of possible values of the random variable $X_{e}\in E$ associated with *e* when *Y* is fixed to its values in outcome *i*. But $H\left(X_{e}|Y\right)\in \left\{0,1\right\}$ and if $H(X_{e}|Y)=0$ then $X_{e}$ is determined by the values of *Y* and |n(i,e,Y)|=1 for all *i*; if $H(X_{e}|Y)=1$ then $X_{e}$ is independent of the values of the variables in *Y* and thus $n\left(i,e,Y\right)=Z_{p}$. This proves that *A* is a secret-sharing matrix. □

Note that this proof remains true for any $p\in N_{\geq 1}$, that is, it does not require the ground set to be a field. The converse of Lemma 3 is true as well: every secret-sharing matroid is *p*-entropic for some *p*. This was observed in Reference [31] and we include a proof for completeness. Together, this observation and Lemma 3 provide an alternative characterization of entropic matroids as secret-sharing matroids.

**Lemma** **4.**
*Every secret-sharing matroid with ground set S is |S|-entropic.*


**Proof.** Let *M* be a secret-sharing matroid and *A* a secret-sharing matrix inducing *M*. Without loss of generality, we may assume that *A* does not contain two identical rows, since this does not affect the structure of the matroid. The definition of secret-sharing matroids implies that the number of rows of *A* is a power ${\left|S\right|}^{r}$ of |S|. We define a probability distribution on the set of random variables $\left\{X_{e}:e\in E\right\}$ by setting the probability that ${\left(X_{e}\right)}_{e\in E}=a$ as ${\left|S\right|}^{-r}$ for every row *a* of *A*.We proceed by induction on |E∖Y| to show that H(Y) (with the Shannon entropy with base |S|) is integral for every Y⊆E and moreover, that the resulting probability distribution on *Y* is the uniform distribution on the distinct rows of A[Y]. This is clearly true for Y=E, since H(E)=r. Let Y⊂E and let e∈E∖Y, then by the induction hypothesis, $H(Y\cup \left\{e\right\})=k\in N$. The matrix $A[Y\cup \left\{e\right\}]$ has ${\left|S\right|}^{k}$ distinct rows and each distinct row has the same probability ${\left|S\right|}^{-k}$. If $H(X_{e}|Y)=0$, then H(Y)=k and distinct rows in $A[Y\cup \left\{e\right\}]$ are distinct rows of A[Y] and thus the distribution of the variables in *Y* is the same as for the variables of $Y\cup \left\{e\right\}$. Therefore, we may assume that fixing the values of the variables in *Y* does not always determine $X_{e}$. This means that n(i,e,Y)=|S| for all *i*. In particular, every distinct row of A[Y] gives rise to |S| distinct rows in $A[Y\cup \left\{e\right\}]$ and thus A[Y] has ${\left|S\right|}^{k-1}$ distinct rows. Each distinct row has the same multiplicity ${\left|S\right|}^{r-k}$ in $A[Y\cup \left\{e\right\}]$ by the induction hypothesis and thus each distinct row of A[Y] has multiplicity ${\left|S\right|}^{r-k+1}$. Now the resulting distribution of the variables in *Y* is a uniform distribution with ${\left|S\right|}^{k-1}$ distinct outcomes, therefore H(Y)=k−1. Clearly, $r_{M}\left(Y\right)=k-1$ and therefore this induction allows us to conclude that the rank in *M* coincides with the entropy of the constructed distribution. This implies the result. □

Seymour [25] proved that the Vamos matroid is not a secret sharing matroid. This implies that it is not an entropic matroid for any *p*.

Moreover, there is a secret-sharing matroid which is not representable over the corresponding field (with |S| elements) and which has been discovered by Simonis and Ashikhmin [32]. This example is the non-Pappus matroid, shown in Figure 1. This matroid has nine elements $\left\{1,\cdots ,9\right\}$ as its ground set *E* and each X⊆E has rank min(|X|,3) with the exception of the eight 3-elements sets shown as colored lines, which each have rank 2. Pappus’ theorem proves that this matroid is not representable over any field.

Simonis and Ashikhmin [32] show that the row space of the matrix
$$\left[\begin{array}{ccccccccc} 10 & 10 & 00 & 10 & 00 & 10 & 10 & 10 & 00\\ 01 & 01 & 00 & 01 & 00 & 01 & 01 & 01 & 00\\ 00 & 00 & 00 & 10 & 10 & 21 & 01 & 10 & 10\\ 00 & 00 & 00 & 02 & 01 & 20 & 12 & 02 & 01\\ 00 & 10 & 10 & 01 & 00 & 01 & 00 & 11 & 10\\ 00 & 01 & 01 & 21 & 00 & 21 & 00 & 10 & 01 \end{array}\right]$$
is a secret-sharing matrix, where each entry of the matrix is considered as an element of $F_{3}^{2}$. They introduce another definition of entropic matroids via codes: a code (subset) $C\subseteq S^{E}$ is *almost affine* if $r\left(Y\right):=\log_{\left|S\right|}\left(\right|C_{Y}\left|\right)\in N_{0}$ for all Y⊆E, where $C_{Y}$ denotes the projection of C to the variables in *Y*. The corresponding matroid *M* with ground set *E* and rank function *r* is called an *almost affine matroid*. It is not hard to see that this definition coincides with secret-sharing matroids by using the codewords in *C* as the rows of the secret-sharing matrix *A* and vice versa. These results show that not all entropic matroids are representable by giving a 9-entropic matroid which is not representable over any field.

## 5. The Case p=2

An $F_{2}$-representable matroid is called *binary*. The goal of this section is to prove the following.

**Theorem** **2.**
*Every 2-entropic matroid is binary.*


To prove this, we use the characterization of binary matroids proved by Tutte [33] stating that a matroid is binary if and only if it has no $U_{2,4}$-minor. $U_{2,4}$ is the uniform matroid of rank two on four elements: E=[4] and F consists of all subsets of *E* of cardinality at most two. Using Tutte’s characterization, the theorem follows from the next lemma.

**Lemma** **5.**
*$U_{2,4}$ is not 2-entropic.*


**Proof.** Suppose for a contradiction that μ is a probability distribution on four random variables $X_{1},\cdots ,X_{4}$ whose entropy is the rank function of $U_{2,4}$, then $H(X_{i})=1$ for all *i* and $H(X_{i},X_{j})=2$ for all i≠j; furthermore $H(X_{1},X_{2},X_{3},X_{4})=2$. This implies that $P\left[X_{i}=a,X_{j}=b\right]=\frac{1}{4}$ for all i≠j and $a,b\in F_{2}$, because the marginal distribution of $X_{i}$ and $X_{j}$ has to be the product of two independent $Ber\left(\frac{1}{2}\right)$ distributions to achieve an entropy of two.Furthermore, $H(X_{i},X_{j}|X_{k},X_{l})=0$ for $\left\{i,j,k,l\right\}=\left[4\right]$ by the chain rule and therefore $P\left[X_{1}=a,X_{2}=b,X_{3}=c,X_{4}=d\right]\in \left\{0,\frac{1}{4}\right\}$ for all a,b,c,d. Without loss of generality, we may assume that $P\left[X_{1}=0,X_{2}=0,X_{3}=0,X_{4}=0\right]=\frac{1}{4}$ but then every other event in which at least two different variables $X_{i}$ and $X_{j}$ are zero must have probability zero, since $P\left[X_{i}=0,X_{j}=0\right]=\frac{1}{4}$. Since $P\left[X_{i}=0,X_{j}=1\right]=\frac{1}{4}$, it follows that all outcomes with three ones have probability $\frac{1}{4}$. Now $\frac{1}{4}=P\left[X_{1}=1,X_{2}=1\right]\geq P\left[X_{1}=1,X_{2}=1,X_{3}=0,X_{4}=1\right]+P\left[X_{1}=1,X_{2}=1,X_{3}=1,X_{4}=0\right]=\frac{1}{2}$, a contradiction. □

## 6. The Case p=3

An $F_{3}$-representable matroid is called *ternary*. The following structure theorem has been proved independently by Seymour [34] and Bixby [35], who attributed it to Reid.

**Theorem** **3**(Seymour [34], Bixby [35]). *A matroid is ternary if and only if it contains no minor isomorphic to $U_{2,5}$, $U_{3,5}$, the Fano plane $F_{7}$ or its dual.*

The Fano plane, shown in Figure 2, has a ground set E=[7] and can be represented over $F_{2}$ by the column vectors of the matrix $\left[\begin{array}{ccccccc} 1 & 1 & 0 & 0 & 0 & 1 & 1\\ 1 & 0 & 1 & 0 & 1 & 0 & 1\\ 1 & 0 & 0 & 1 & 1 & 1 & 0 \end{array}\right]$, that is, a set is independent if and only if it contains at most three vectors and it does not contain all three vectors on any line (including the circle).

**Lemma** **6.**
*$U_{2,5}$ is not 3-entropic.*


**Proof.** Suppose for a contradiction that there exist $X=(X_{1},\cdots ,X_{5})$ such that $H\left(A\right)=\min\left\{|A|,2\right\}$ for all $A\subseteq \left\{X_{1},\cdots ,X_{5}\right\}$. Then, for any choice of $\left\{a,b,c,d,e\right\}=\left\{1,2,3,4,5\right\}$, we have that $H(X_{a},X_{b},X_{c}|X_{d},X_{e})=0$ and thus for any vector $x\in F_{3}^{5}$,
$$P\left[X_{a}=x_{a},X_{b}=x_{b},X_{c}=x_{c}|X_{d}=x_{d},X_{e}=x_{e}\right]\in \left\{0,1\right\}$$
and $P\left[X=x\right]\in \left\{0,\frac{1}{9}\right\}$.As in the proof for $U_{2,4}$, we may assume that $P\left[X=0\right]=\frac{1}{9}$ but then any other event with at least two zeros must have probability 0. This leaves six events, five with one zero and one with no zeros; but each of them has probability at most $\frac{1}{9}$, thus the total probabilities add up to at most $\frac{7}{9}$, a contradiction. □

**Lemma** **7.**
*$U_{3,5}$ is not 3-entropic.*


**Proof.** As before, we suppose for a contradiction that there is a vector $X=(X_{1},\cdots ,X_{5})$ of random variables such that $H\left(A\right)=\min\left\{|A|,3\right\}$ for all $A\subseteq \left\{X_{1},\cdots ,X_{5}\right\}$.Every three distinct variables are independent and they determine the other two variables. It follows that, for every event, its probability is either zero or $\frac{1}{27}$. But there are only 81 outcomes and 27 of them occur with positive probability. Each of those 27 must differ from the others in at least three places, because if two outcomes are equal in three positions, the other two are determined and thus equal. This means that the Hamming balls of radius 1 around the outcomes with positive probability are disjoint. Each of these Hamming balls contains 11 elements: the outcome with positive probability and the outcomes in which one variable is flipped to one of the two other possible values. Therefore, we have at least 27×11=297 outcomes, a contradiction. □

**Lemma** **8.**
*The Fano plane is not 3-entropic.*


**Proof.** Suppose for a contradiction that the Fano plane is 3-entropic and that $X=\left\{X_{1},\cdots ,X_{7}\right\}$ is a set of random variables whose entropy corresponds to their rank in the Fano matroid as shown in Figure 2. Since the maximum size of an independent set in the Fano matroid is three, any three independent variables determine the values of all the others; in particular, there are at most 27 outcomes with positive probability, which we denote by their values on the independent set $X_{1},X_{2},X_{3}$. Since $H(X_{1},X_{2},X_{3})=3$, each of these outcomes has probability $\frac{1}{27}$, whereas all other outcomes have probability zero. It follows that we have a map $f:F_{3}^{3}\to F_{3}^{4}$ mapping the values on $X_{1},X_{2},X_{3}$ to the values on $X_{4},\cdots ,X_{7}$, where $X_{2}$ and $X_{3}$ determine $X_{7}$, $X_{1}$ and $X_{2}$ determine $X_{5}$ and $X_{3}$ and $X_{1}$ determine $X_{6}$ but every change of one of $X_{1},X_{2},X_{3}$ must change $X_{4}$.We consider the set of nine assignments of $X_{1},X_{2},X_{3}$ for which $X_{4}=0$. If every two of these have pairwise distance at least three, we can only have three distinct assignments. This implies that we may assume that there are two assignments with distance two. Furthermore, if we fix any two digits, exactly one choice is valid for the remaining digit. Therefore, up to isomorphism (exchanging symbols), the set looks as follows: $\left\{000,012,021,102,111,120,201,210,222\right\}$; and thus $X_{4}=X_{1}+X_{2}+X_{3}$.The random variables $X_{2},X_{3},X_{4}$ determine $X_{5},X_{6},X_{7}$ and $X_{1}$. In particular, both of the pairs $X_{1}$, $X_{1}+X_{2}+X_{3}$ and $X_{2}$, $X_{3}$ determine $X_{7}$.Changing $X_{1}$ does not change $X_{7}$ and neither does simultaneously adding *k* to $X_{2}$ and subtracting *k* from $X_{3}$. Therefore, keeping $X_{2}+X_{3}$ constant will keep $X_{7}$ constant and $H(X_{7}|X_{2}+X_{3})=0$, and $H(X_{2}+X_{3}|X_{7})=0$. This implies that there is a one-to-one correspondence between $X_{7}$ and $X_{2}+X_{3}$ and similarly between $X_{6}$ and $X_{2}+X_{4}$ and between $X_{5}$ and $X_{3}+X_{4}$. But then $X_{5},X_{6},X_{7}$ allow us to find $X_{2}+X_{3}$, $X_{2}+X_{4}$ and $X_{3}+X_{4}$ and thus $2X_{2}+2X_{3}+2X_{4}$ and $X_{2}+X_{3}+X_{4}$ (since 2≠0 in $F_{3}$), which is $X_{1}$. This shows that $H(X_{1}|X_{5},X_{6},X_{7})=0$ and thus $3=H\left(X_{1},X_{5},X_{6},X_{7}\right)=H\left(X_{5},X_{6},X_{7}\right)=2$, a contradiction. □

The above proof actually shows that the Fano plane is not *p*-entropic for any p>2, which gives an alternative proof that it is not $F_{p}$-representable for p>2 either.

The dual $F_{7}^{*}$ of the Fano plane is $F_{2}$-representable and a representation is given by the columns of the matrix $\left[\begin{array}{ccccccc} 1 & 1 & 1 & 1 & 1 & 1 & 1\\ 1 & 1 & 0 & 0 & 0 & 1 & 1\\ 1 & 0 & 1 & 0 & 1 & 0 & 1\\ 1 & 0 & 0 & 1 & 1 & 1 & 0 \end{array}\right].$ This shows that every 3-element set is independent in $F_{7}^{*}$, thus its circuits are exactly the complements of the three-element circuits of the Fano plane. To give a better understanding of these matroids, we expanded the symmetrical representation of $F_{7}$ given in Reference [36] and shown in Figure 3a to $F_{7}^{*}$. The result is shown in Figure 3b. Each color connects the elements of a circuit in one figure and the corresponding circuit given by its complement in the other figure. The cyclical order of the nodes in Figure 3a yields a rainbow Hamilton cycle (one edge of each color) in Figure 2.

**Lemma** **9.**
*The dual of the Fano plane is not 3-entropic.*


**Proof.** Suppose for a contradiction that $X=(X_{1},\cdots ,X_{7})$ is a vector of random variables whose entropy coincides with the rank function of $F_{7}^{*}$. Since $H(X_{2},X_{3},X_{4},X_{5})=4$ and H(X)=4, $P\left[X=x\right]\in \left\{0,\frac{1}{81}\right\}$ for all $x\in F_{3}^{7}$. We refer to the events with positive probability as outcomes.By permuting the symbols, we may assume that 0000000 is a possible outcome. We consider the other outcomes $\left(X_{1},X_{6},X_{7}\right)$ for $X_{2}=0$. No two of these outcomes can have distance one, because $X_{1},X_{2},X_{6},X_{7}$ is a cycle, so for fixed $X_{2}$, any two distinct possible outcomes must have distance at least two on their restriction to $\left(X_{1},X_{6},X_{7}\right)$. In the proof of the previous lemma, we have already to shown that by switching digits, we may assume that the set of images is $\left\{000,012,021,102,111,120,201,210,222\right\}$. As shown in Figure 4, this also determines the other two sets (but not necessarily which of them is which). This shows that $X_{1}+X_{6}+X_{7}$ is sufficient to determine $X_{2}$ and vice versa; by flipping symbols 1 and 2 for $X_{2}$, we may assume that $X_{1}+X_{6}+X_{7}=X_{2}$.We now fix $X_{3}$. Then $X_{4}$ is determined by either $X_{1},X_{2}=X_{1}+X_{6}+X_{7}$ or $X_{6},X_{7}$ and thus changing $X_{1}$ or adding *k* to $X_{6}$ and subtracting it from $X_{7}$ does not change $X_{4}$. This implies that $X_{4}$ depends only on $X_{6}+X_{7}$ (and $X_{3}$) and thus $H(X_{3},X_{4},X_{6}+X_{7})=2$. Analogously, $H(X_{3},X_{5},X_{1}+X_{7})=2$ and $H(X_{4},X_{5},X_{1}+X_{6})=2$. Therefore, $X_{3}$, $X_{4}$ and $X_{5}$ determine $X_{6}+X_{7}+X_{1}+X_{7}+X_{1}+X_{6}=2\left(X_{1}+X_{6}+X_{7}\right)=2X_{2}$ and since 2≠0 in $F_{3}$, this shows that $H(X_{2},X_{3},X_{4},X_{5})=3$, contradicting the assumption that *X* had the entropy function given by the rank in $F_{7}^{*}$. □

Combining these four lemmas with the characterization of ternary matroids, we have proved the following theorem (the interesting part being the only if part).

**Theorem** **4.**
*A matroid is 3-entropic if and only if it is $F_{3}$-representable.*


## 7. Comments for General Primes p

For ground sets of arbitrary size *p*, being *p*-representable is a stronger assumption than being *p*-entropic as the example of Simonis and Ashikhmin [32] of the non-Pappus matroid (see Figure 1) shows. However, no counterexamples exist in the case where the ground set has prime order.

In this section, we show that for primes *p*, every *p*-entropic matroid of rank at most two is linear, that is, let *M* be an entropic matroid with ground set *E* and H(E)≤2, then *M* is linear. If H(E)<2, this is true since any basis has at most one element. Furthermore, we may assume that every X∈E satisfies H(X)=1, for otherwise *X* is deterministic and is represented by the zero vector in every linear representation.

**Lemma** **10.**
*Let M be a p-entropic matroid of rank 2. If there are two elements X and Y in the ground set E with H(X,Y)=1, then M is $F_{p}$-linear if and only if $M\setminus \left\{X\right\}$ is.*


**Proof.** If *M* is $F_{p}$-representable, then so is $M\setminus \left\{X\right\}$, since it is a minor-closed property. Suppose that $M\setminus \left\{X\right\}$ is representable and let $f:E\setminus \left\{X\right\}\to V$ be a representation and let g:E→V be defined as f(Z) for Z≠X and f(X)=f(Y). Let S⊆E. Then dim(span(g(S)))=H(S) for X∉E. If X∈S but Y∉S, then $\dim\left(span\left(g\left(S\right)\right)\right)=\dim\left(span\left(f\left(S\cup \left\{Y\right\}\right)\right)\right)=H\left(S\cup \left\{Y\right\}\right)$ and
$$\begin{array}{cc} H(S\cup \left\{Y\right\}) & =H(S)+H(Y|S)\\ \; & =H(S)+H(X|S)+H(X,Y|S)-H(X|S)+H(Y|S)-H(X,Y|S)\\ \; & =H\left(S\right)+H\left(X|S\right)+H\left(Y|X,S\right)-H\left(X|Y,S\right)=H\left(S\cup \left\{X\right\}\right). \end{array}$$If X,Y∈S, then $\dim\left(span\left(g\left(S\right)\right)\right)=\dim\left(span\left(f\left(S\setminus \left\{X\right\}\right)\right)\right)=H\left(S\setminus \left\{X\right\}\right)=H\left(S\right)$ by applying submodularity to the sets $\left\{X,Y\right\}$ and $S\setminus \left\{X\right\}$. This proves that *g* is an $F_{p}$-representation of *M*. □

With the above lemma, we have reduced the problem to considering uniform matroids. For any prime *p*, the uniform matroid $U_{2,p+1}$ is $F_{p}$-representable by choosing the images of *E* as
$$\left(0,1\right),\left(1,0\right),\left(1,1\right),\left(1,2\right),\cdots ,\left(1,p-1\right)\in F_{p}^{2}.$$

Each pair of these p+1 vectors is independent and a basis of $F_{p}^{2}$, thus they represent $U_{2,p+1}$. The following lemma shows that any larger uniform matroid is neither *p*-entropic nor $F_{p}^{2}$-representable.

**Lemma** **11.**
*The uniform matroid $U_{2,p+2}$ is not p-entropic for any $p\in N_{\geq 2}$.*


**Proof.** Suppose not and let *C* denote the set of possible outcomes for a probability distribution on p+2 variables representing $U_{2,p+2}$. By changing symbols, we may assume that (0,⋯,0) is a possible outcome. Furthermore, there are $p^{2}$ outcomes and hence *p* of them begin with a zero. These *p* outcomes have the same value at the first coordinate $X_{1}$ but all other values are distinct (i.e., each $X_{i}$ for i>1 takes all of its *p* possible values exactly once among these *p* outcomes, including value zero for outcome (0,⋯,0)). Therefore, we can simultaneously change the other symbols so that these p outcomes become (0,0,⋯,0),(0,1,⋯,1),(0,2,⋯,2),⋯,(0,p−1,⋯,p−1). But then any other outcome not starting with zero satisfies that $X_{2},\cdots ,X_{p+2}$ all take different values in $Z_{p}$. Since there are only *p* values but p+1 variables, this is a contradiction. □

This shows that line matroids, which are among the forbidden minors of binary and ternary matroids, are *p*-entropic if and only if they are $F_{p}$-linear.

## 8. Application: Entropic Matroids in Coding

We recall here a result proved in Reference [12] that makes entropic matroids emerge in a probabilistic context and which gives further motivations to studying entropic matroids. The result gives in particular a rate-optimal code for compressing correlated sources, similarly to the channel counter-part developed in Reference [37].

Let $X^{n}=\left(X_{1},\cdots ,X_{n}\right)$ be an i.i.d. sequence of discrete random variables taking values in $X^{m}$. That is, $X^{n}$ is an m×n random matrix with i.i.d. columns of distribution μ on $X^{m}$. One can assume that the support of X is finite (countable supports can be handled with truncation arguments) and to further simplify, we assume that X is binary, associating each element in the binary field, that is, X=GF(2).

Due to the i.i.d. nature of the sequence, the entropy of $X^{n}$ is the sum of each components’ entropies H(μ), i.e.,
(7)$$\begin{array}{c} H\left(X^{n}\right)=nH\left(\mu \right). \end{array}$$

The next result shows that it is possible to transform the sequence $X^{n}$ with an invertible map that extracts the entropy in subsets of the components. In words, the transformation takes the i.i.d. vectors under an arbitrary μ to a sequence of distributions that correspond in the limit to entropic matroids.

**Theorem** **5**(Abbe [12]). *Let m be a positive integer, n be a power of 2 and $X^{n}$ be an m×n random matrix with i.i.d. columns of distribution μ on $F_{2}^{m}$. Let $Y^{n}=X^{n}G_{n}$ over $F_{2}$, where $G_{n}={\left[\begin{array}{cc} 1 & 0\\ 1 & 1 \end{array}\right]}^{⊗\log_{2}\left(n\right)}$. For any $\varepsilon =O(2^{-n^{\beta }})$, β<1/2, we have*
(8)$$\begin{array}{c} |\{i\in \left[n\right]:H\left(Y_{i}\left[S\right]|Y^{i-1}\right)\notin Z\pm \varepsilon ,\;\mathrm{for}\;\mathrm{any}\;S\subseteq \left[m\right]\}|=o\left(n\right). \end{array}$$

In other words, one starts with an i.i.d. sequence of random vectors under a distribution μ that defines an *entropic polymatroid*[m]⊇S↦H(S) and after the transformation $G_{n}$, one obtains a sequence of random vectors which is no longer i.i.d. but where each random vector given the past defines an *entropic matroid* in the limit. Having a matroid structure is of course much easier to handle for compression purposes, one simply has to pick a basis for each matroid, store the components in that basis and the other components are fully dependent on these so they can be recovered without being stored. Of course, in practice *n* is large but finite, and each random vector defines a polymatroid that is *close* to a matroid but a continuity argument allows to show that the components outside of the bases can still be recovered but only *with high probability*. Since a compression code is allowed to fail with a low probability of error, this is not an issue. Understanding the structure of these entropic matroids allows then one to better understand how the stored components can be allocated over the different components—see Reference [12] for further details.

## Figures and Tables

**Figure 1 entropy-21-00948-f001:**
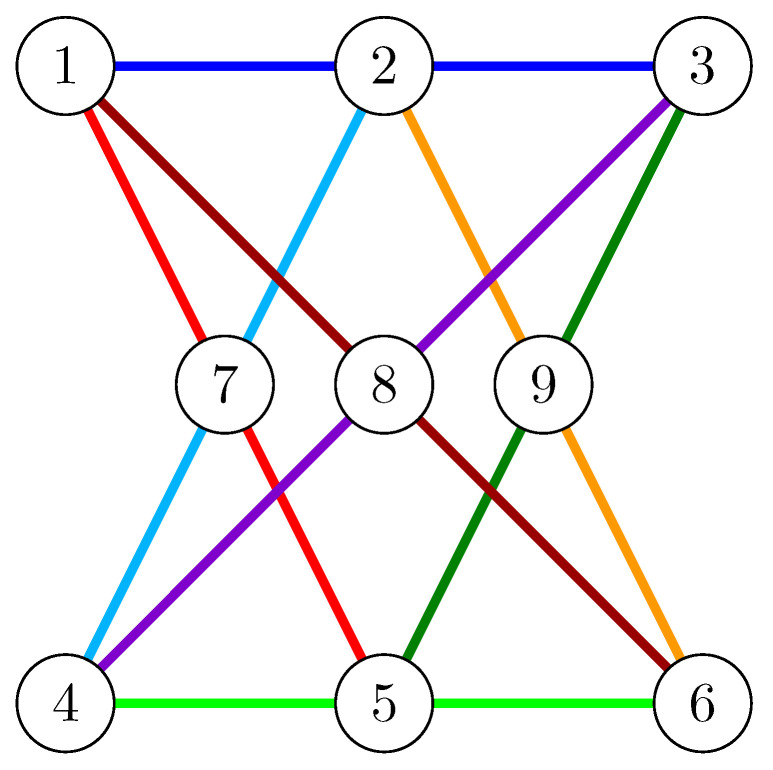
The non-Pappus matroid.

**Figure 2 entropy-21-00948-f002:**
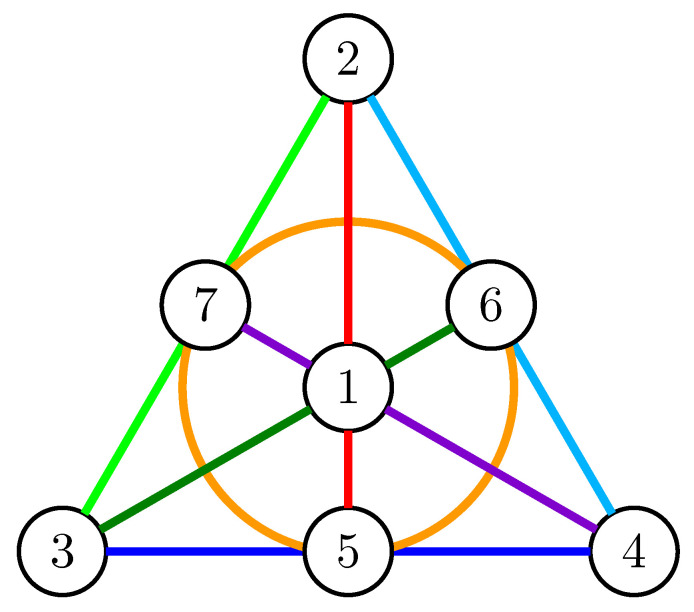
The Fano plane.

**Figure 3 entropy-21-00948-f003:**
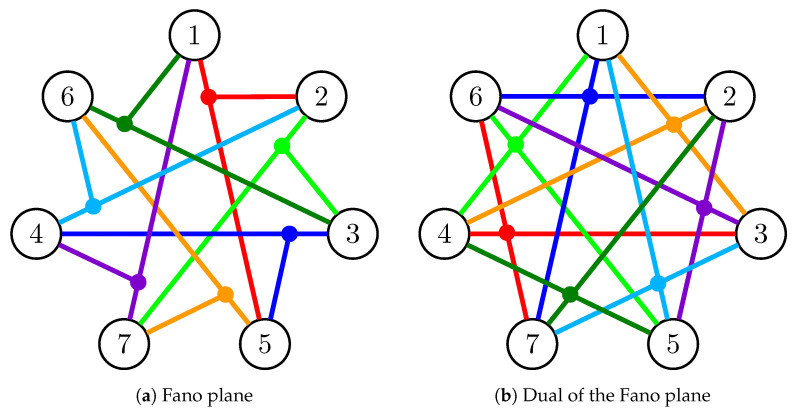
A symmetrical view of the circuits of the Fano plane and its dual.

**Figure 4 entropy-21-00948-f004:**
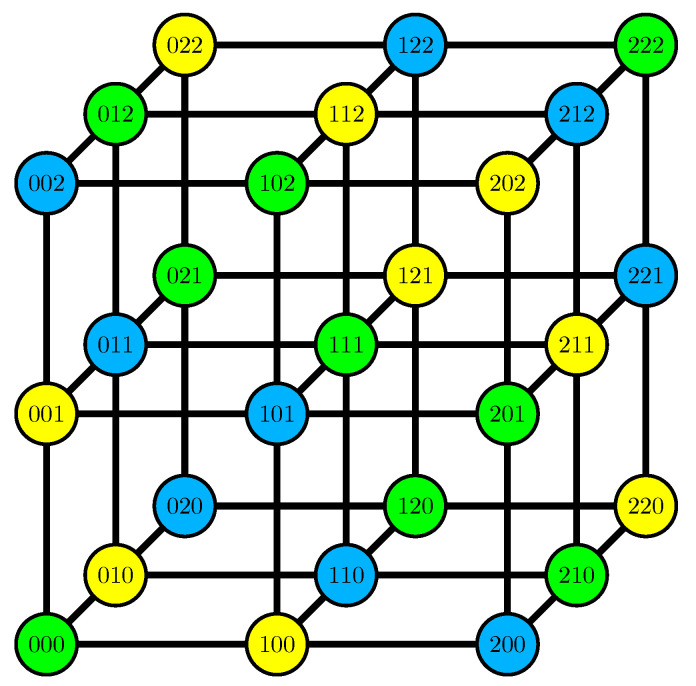
Values of $\left(X_{1},X_{6},X_{7}\right)$ colored by corresponding value of $X_{2}$.

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
