# Peer review of "Entropic Matroids and Their Representation"

_entropy, 2019, doi:10.3390/e21100948_

Round 1

Reviewer 1 Report

Matroids in general is a class of rather abstract combinatorial ojbjects. Traditionally, mathematicians are interested in representing a matroid concretely by linear algebra over a fixed field. In particular, on is interested whether a matroid an be represented by a matrix over a finite field. Another representation of matroid is to use the concept of entropy from information theory. Such a matroid is called entropic matroid.

The authors make a relationship between these two different ways of representing a matroid. By "p-entropic", it means that the underlying sample space of the random variables has cardinality p. In my opinion, the main contribution of this paper is to prove that every 2-entropic matroid is linearly representable over GF(2), the finite field of size 2 (Theorem 2), and every 3-entropic matroid is linearly representable over GF(3) and vice versa (Theorem 4). The proofs invoke some known results from matroid theory.

The first part of the paper also contains some result about the relationship between p-entropic matroid and secret-sharing matroid.

The presentation of the paper is good in general. However, I feel that the content of the last two sections (Sections 6 and 7) are not very mature. I would suggest the authors to consider removing the last two sections, and concentrate on the proofs of Lemmas 3 and 4, and Theorems 2 and 4, which are the main results of this paper.

Author Response

We are thankful to the reviewers for their useful reviews.

R1: we agree that section 6 is not as mature, but also would like to keep the partial result to motivate potentially future work on the prime case. We changed  the section name to `Comments for general primes' to stress its partial content. Section 7 is not a contribution of the paper but a motivation to studying q-entropic matroids (and in fact how they emerge in a concrete application) so it seems useful. We have now better stressed this to avoid confusion.

Reviewer 2 Report

See attached file.

Author Response

We are thankful to the reviewers for their useful reviews.

R2: we are grateful for the literature pointers. We were not familiar with all components of this literature, so this is helpful. We were aware of some of the applications of entropic polymatroids and had not cited these as they were not directly related to the paper; we have now added them. 

Regarding point 3; unfortunately we do not see a connection. We will however add a comment about the well-studied line of work driven by the ground set cardinality restriction.

Other comments have been addressed in the revision; see some responses below.  Thank you for these too.

"Something is confusing in the definition of contraction. Are you sure you meant A F?"

Yes -- otherwise B union A in F is never true. 

"Proof of Lemma 2: end of page 4, two cases where X should be Xi, I think. In any case, it is obvious that conditional distributions are legitimate distributions, hence have entropies which satisfy the required constraints (submodularity, etc.). The work in this proof is focused on the integer values. This should be clarified. Maybe, give an outline before starting the proof."

We corrected X to Xi. We added an outline: the goal is to show that the probability distribution we construct has as its entropy the rank of the matroid.

Round 2

Reviewer 2 Report

All my comments were addressed. Thank you. One tiny thing: line 167, I believe the . should be a ,

Thank you.